# Perceived symptoms as the primary indicators for 30-day heart failure readmission

Kelley M. Anderson[1]*, Dottie Murphy[2], Hunter Groninger[3,4], Paul Kolm[5], Haijun Wang[5], Vera Barton-Maxwel[6]

1 Department of Professional Nursing Practice, Georgetown University School of Nursing & Health Studies, Washington, District of Columbia, United States of America, 2 Department of On-line Nursing, Liberty University, Lynchburg, Virginia, United States of America, 3 Division of Palliative Care, Medstar, Washington Hospital Center, Washington, District of Columbia, United States of America, 4 School of Medicine, Georgetown University, Washington, District of Columbia, United States of America, 5 Division of Bioinformatics and Biostatistics, MedStar, Health Research Institute, Hyattsville, Maryland, United States of America, 6 Department of Advanced Nursing Practice, Georgetown University School of Nursing & Health Studies, Washington, District of Columbia, United States of America

⊕ These authors contributed equally to this work.
* kma25@georgetown.edu

## Abstract

### Background

To identify 30-day rehospitalizations in patients discharged with heart failure (HF) based on clinical indications, physiologic measures and symptoms.

### Methods

Fifty-six patients with heart failure participated. After discharge to home, clinical indicators of dyspnea, fatigue, orthopnea, dyspnea with exertion, daily weight, edema, heart rate, blood pressure, mental condition, medication adherence, and overall well-being were reported by participants daily for up to 30 days.

### Results

Joint modeling of longitudinal and time-to-event approach was applied to assess the association of readmission with longitudinal measurements. There was no association between demographic, physiological, or laboratory variables and re-hospitalization within 30 days post discharge. Perceptions of dyspnea ($p$ = .012) and feeling unwell ($p$ < .001) were associated with rehospitalization. Patients struggling to breath were 10.7 times more likely to be readmitted than those not struggling to breath.

### Conclusion

Perceived measures, particularly dyspnea and feeling unwell were more important factors than demographic, physiological, or laboratory parameters in predicting 30-day

**Data Availability Statement:** The data underlying the results presented in the study are available from the corresponding author and study

statistician, and attached as a Supporting information file.

**Funding:** KMA: This project has been funded in whole or in part with Federal funds (UL1TR000101 previously UL1RR031975) from the National Center for Advancing Translational Sciences (NCATS), National Institutes of Health, through the Clinical and Translational Science Awards Program (CTSA), a trademark of DHHS, part of the Roadmap Initiative, Re-Engineering the Clinical Research Enterprise. The sponsor or funder did not play any role in study design, data collection and analysis, decision to publish, or preparation of the manuscript.

**Competing interests:** NO authous have competing interests.

rehospitalizations in this racially diverse cohort. The symptomatic experience of heart failure is an important indicator of rehospitalization.

## Introduction

According to the universal definition, heart failure (HF) is a complex clinical syndrome with current or prior symptoms or signs, corroborated by objective measures [1]. Signs and symptoms of HF are due to structural and/or functional cardiac abnormalities and typically include fatigue, reduced exercise tolerance, edema, and various presentations of dyspnea [1]. The worldwide prevalence of HF is estimated at 64 million people, including 6.2 million adults in the United States (US) [2–4]. Hospitalizations for HF in the US are estimated at over 800,000 per year with a readmission rate of approximately 20% at 30 days and up to 50% six months after discharge [5, 6]. Hospitalizations remain markers of disease severity and portend future rehospitalizations and mortality.

Demographic, historical, laboratory, clinical, functional, and physiologic predictors of HF outcomes, including readmissions have been previously evaluated. Investigated predictors included demographic and health history such as age [7–9], sex [7, 10], ethnicity [10], body mass index [11], previous hospitalizations [12, 13], ischemic etiology [12–14], New York Heart Association functional class, comorbid conditions [14–17], chronic use of loop diuretics [15], and cardiac consultation [18]. Other identified predictors of HF outcomes included clinical, physiologic, and laboratory findings such as residual congestion, left ventricular ejection fraction (LVEF), left atrial and ventricular enlargement, brain natriuretic peptide, elevated creatinine, and anemia [19–22]. However, despite decades of research to identify predictors of readmission, rehospitalization continue to be significant in the HF population [6, 23]. As Gherorghiade stated, "improving discharge outcomes is the single most important goal in the management of acute HF syndromes" [24].

In addition to hospitalizations, many patients with HF suffer from substantial symptom burden, predominately dyspnea, that remains the primary cause of recurrent rehospitalizations and emergency department (ED) visits [20, 25]. Dyspnea is the primary presenting symptom to the ED in patients with acute decompensated HF and often results in hospital admission [25, 26]. Although many factors contribute to decompensated HF, the reason patients seek care is the experience of symptoms, specifically dyspnea.

The UCSF Symptom Management Theory (SMT) describes that symptom recognition, symptom response, and outcomes are complex, multifaceted, and interrelated [27, 28]. The SMT describes symptom management as a multi-dimensional process with the premise that symptom experience, symptom management, and symptom outcomes are interrelated within the context of person, environment, and health and illness [28]. In this model, a symptom is defined "as a subjective experience reflecting changes in a person's biopsychosocial function, sensation or cognition;" with symptom perception the "conscious cognitive interpretation of information gathered by the senses in the context of a particular environment or situation" [27, 29].

Symptoms that negatively impact function or well-being with sufficient frequency or severity result in help-seeking responses, including accessing the ED and acute care setting [30, 31]. However, studies evaluating the subjective post-hospital experience of patient-reported symptoms to predict HF readmissions are lacking. This study addresses this gap by investigating daily symptomatic, physiologic, and medication adherence characteristics and the relationship

of these patient-centric experiences to expand our understanding of factors contributing to 30-day HF post-hospitalization readmissions.

## Methods

We conducted a prospective, observational pilot study. Georgetown University Institutional Review Board approved the study (IRB #2015–0336), and written informed consent was obtained from participants. Inclusion criteria included: 1) at least 50 years of age; 2) admitted to the hospital for at least 24 hours, with the primary diagnosis of HF; 3) able to read and write in English; and 4) discharged to the home environment. Participants were excluded if: 1) a cardiac transplant candidate; 2) experienced an acute coronary event within the previous 30 days of index hospitalization; 3) experienced percutaneous coronary intervention or coronary artery bypass grafting within the previous 30 days of index hospitalization; 4) had end-stage renal disease/hemodialysis 5) had a left ventricular assist device, 6) weighed more than 400lbs, or 7) were unable to stand for 90 seconds independently. Enrollment reflected the gender and ethnic groups of the patient population at the study institution. The setting included all HF inpatient units in a large, academic health center in the mid-Atlantic region of the US. The readmission rates at this institution are congruent with those reported at a national level.

Participants admitted with a diagnosis of HF were identified by nurses on the HF units. Once clinically stable, participants were preliminarily screened to determine study eligibility based on inclusion and exclusion criteria. Following informed consent and enrollment, baseline demographic information and medical histories were abstracted from the electronic health records and participant reports. At the end of hospitalization, intake and output measures, physiologic measures, laboratory findings, and medications were abstracted from the electronic health record. Participants were educated on daily monitoring and measurements of weight, pulse and blood pressure. If needed, scales and blood pressure cuffs were provided and their use was demonstrated. The self-monitoring process was derived from the existing literature, including the literature on HF action plans that are used for self-care management and symptom recognition [32–35] and consistent with key data elements and definitions for HF [36, 37]. The 12-item self-monitoring process was divided into 4 components: physiologic measures (HR, BP, weight), medication compliance (yes/no), HF signs and symptoms (yes/no or scale 1–10), and one question regarding visits to the ED or hospitalization. Participants completed the form beginning the day after discharge. For example, if the patient returned home on Monday, day one began on Tuesday and continued for a total of 30 days, or until readmission.

### Statistical methods

Analyses were performed to evaluate the participant sample, primary hypothesis and specific aims. After completing a descriptive evaluation of the study data, univariate Cox proportional hazard model was used to identify baseline key risk factors such as demographics and laboratory measures. A Kaplan Meier plot was conducted to demonstrate the probability of readmission over time. Joint modeling of longitudinal and time-to-event approach was applied to assess the association of readmission with longitudinal measurements including objective measures such as weight, blood pressure, heart rate and survey questions such as breathing, edema and tiredness. Longitudinal model and time-to event model were connected through random effect of longitudinal data, and current value method was used as shared parameter. % JM macro was used to fit the model [38].

**Missing data.** Missing baseline continuous data (e.g., age, BNP) were imputed using predictive mean matching (PMM) [39]. Missing values were imputed by logistic regression for

dichotomous variables. Five sets of complete data were generated for readmission analysis. Analyses were completed for each complete data set and then combined for estimation of overall results. Missing responses for patient daily questions were "filled in" by last value carried forward (LVCF) over the 30-day period.

**Variable selection.** As there was a large number of potential predictors of readmission, penalized regression via elastic-net methodology was used to identify the most important predictors [40]. Elastic-net penalizes regression coefficients towards zero so that only the important predictors remain. There were 43 demographic, clinical, and laboratory variables included in the elastic-net analysis (Table 1). This methodology was applied to each of the five imputed data sets. For each of the 5 data sets, variables were ranked according to their order of selection. Predictors that appeared in all five data sets were then used in a Cox proportional hazards regression model of readmission.

**Self-monitoring.** The relationship between readmission and answers to questions 1–13 over the 30 days after discharge was analyzed by joint modeling of longitudinal and readmission data [38, 41]. Readmission-free days were modeled as exponential curves. If the joint modeling method failed to converge, Poisson regression, with time to censoring or readmission as the exposure, was used to model readmission. Baseline variables that were predictors of readmission were included in the joint modeling analyses along with age and sex even if the latter two were not identified in the elastic-net analysis as important predictors. Longitudinal trajectory plots were made to assess the joint relationship between question responses over 30 days and readmission. In these plots, the time scale was adjusted such that 0 represented the point of readmission or censoring. Separate plots were made for patients who were readmitted and those who were not readmitted within 30 days (censored). A Kaplan-Meier readmission curve was constructed for patients with follow-up data. All analyses were performed using SAS 9.4 or Stata v.16 (College Station, TX).

## Ethical aspects of the research

Study approval was obtained from the University and health care facility Institutional Review Boards prior to initiation of the study. Participants who completed the study received a gift card of $35 for their time and effort.

## Results

### Sample characteristics

Table 1 presents descriptive statistics for baseline variables and participant characteristics. A total of 56 participants were included in this study with a mean age of 67 years (SD 10.8). Females comprised 54% of the participants, and the majority were Black (63%), approximately half of the participants (45%) were married, and the majority (89%) had a previous history of HF. Comorbidities included hypertension (76%), diabetes mellitus (51%), valvular heart disease (49%), chronic kidney disease (48%), atrial fibrillation (46%), pulmonary disease (37%), and ischemic heart disease (34%). The majority of participants (84%) were not actively using tobacco. The mean LVEF was 33% (SD16.7) with 66% of participants characterized as non-ischemic HF. Of the 56 enrolled patients, 31 (55.4%) completed the 30-day follow-up, with 32.3% readmissions within 30 days of discharge.

### Clinical data

Mean physiologic parameters were evaluated including heart rate: 83 bpm (SD 16), blood pressure 116/65 mmHg (SD 18.8/8.3), and BMI 31 kg/m$^2$ (SD 8.3). Mean laboratory values

**Table 1. Participant characteristics (n = 56).**

| Variable | Statistics |
|---|---|
| Baseline Characteristics | |
| Age (years), Mean | 66.6 ± 10.8 |
| Male (%) | 26 (46.4) |
| **Race** | |
| Black (%) | 35 (62.5) |
| White (%) | 20 (35.7) |
| Other (%) | 1 (1.8%) |
| **Marital Status** | |
| Married (%) | 25 (44.6) |
| Single (%) | 19 (33.9) |
| Widowed (%) | 5 (8.9) |
| Divorced (%) | 7 (12.5) |
| History of Heart Failure (%) | 49 (89.1) |
| Clinical Data | Mean (SD) |
| NT-proBNP, pg/mL | 8676 (14223) |
| Sodium, mEq/L | 138 (4.6) |
| BUN, mg/dL | 33 (16.9) |
| Creatinine, mg/dL | 1.47 (0.7) |
| Hemoglobin, g/dL | 11.9 (1.9) |
| Left ventricular ejection fraction, % | 33 (16.7) |
| Heart rate, bpm | 83 (15.8) |
| Systolic blood pressure, mmHg | 116 (18.8) |
| Diastolic blood pressure, mm Hg | 65 (8.3) |
| Weight at admission, kg | 91 (27.7) |
| Weight at discharge, kg | 86 (26.2) |
| Body Mass Index, kg/m$^2$ | 31 (8.3) |
| Intake/Output during hospitalization | -796 (955) |
| Dyspnea, 0–10 scale | 7.8 (1.7) |
| Lower extremity edema, Mean (%) | 25 (46.3) |
| **Comorbidities, n(%)** | |
| Hypertension | 42 (76.4) |
| Diabetes | 27 (50.9) |
| Coronary Heart Disease | 18 (34.0) |
| Valvular Heart Disease | 25 (49.0) |
| Atrial fibrillation | 25 (46.3) |
| Chronic kidney disease | 25 (48.1) |
| Pulmonary disease | 19 (36.5) |
| **Habits, n (%)** | |
| Tobacco | 9 (16.1) |
| Alcohol | 17 (30.9) |
| **Discharge medications, n (%)** | |
| ACE inhibitor/Angiotensin receptor blockers | 28 (52.8) |
| Beta-blockers | 32 (60.4) |
| Diuretic | 45 (83.3) |
| Digoxin | 8 (14.6) |
| Spironolatone | 14 (26.4) |
| Statin | 29 (52.7) |

*(Continued)*

**Table 1.** (Continued)

| Variable | Statistics |
|---|---|
| Aspirin | 29 (52.7) |
| **Independently performed, n (%)** | |
| Ambulation | 50 (89.3) |
| Bathing | 49 (89.1) |
| Toileting | 50 (90.9) |
| ADLs | 50 (90.9) |
| Assisted Devices | 11 (20) |

Statistics are mean ± one standard deviation for continuous variables, n (%) for categorical variables.

included: NT-pro BNP 8676 pg/dL (SD 14222.7), serum sodium 138 mEq/L (SD 4.6), creatinine 1.47 mg/dL (SD 0.70), BUN 33.16 mg/dL (SD 16.9), and hemoglobin 11.9 g/dL (SD 1.9). Participants were prescribed the following medications at discharge: ACEi or ARB (53%), beta-adrenergic blocker (60%), diuretic (83%), spironolactone (26%), and digoxin (15%). Mean urine output loss during hospitalization was 796 ml (SD 955, range +1615 to -2788). Prior to hospital discharge, participants' mean rating of their ease of breathing was 7.8 (SD 1.7) on a Likert scale of 1–10 (1 being the worst, 10 being the best). Lower extremity edema was present in 46% of participants at the end of hospitalization. At the time of discharge, the majority (91%) of participants were able to perform activities of daily living (ADLs) including ambulation (89%), bathing (89%) and toileting (91%) without assistance.

## Readmission factors

A univariate Cox regression analysis was used to identify risk factors for readmission (Table 2). There were no significant differences in 30-day readmission rates related to baseline parameters including sex, age, ethnicity, marital status, history of HF comorbidities, HF etiology, tobacco or alcohol use, laboratory values, BMI, medications, ADLs, LVEF, heart rate, blood pressure, urine loss, lower extremity edema or dyspnea. The only variable from the elastic-net analysis associated with readmission was statin medications at discharge. Age and sex were not identified by the analysis but were included as covariates in subsequent Cox models. Adjusted for age and sex, patients discharged on statins were 10 times more likely to be readmitted than those not discharged on statins (Hazard Ratio = 9.85, 95% CI [1.25–77.97], $p$ = .026).

**Joint models.** Fig 1 includes longitudinal trajectory plots for questions 1 (struggling to breath) and 13 (feeling in general). The plot for those readmitted indicate an exponential increase in struggling to breath as the readmission date approached and a linear decrease in how patients felt that began approximately two weeks before readmission. For those not readmitted (censored), there was virtually no longitudinal change for these questions. Table 3 presents hazard ratios and 95% confidence intervals for each of the questions. Perceptions of dyspnea ($p$ = .012) and feeling unwell ($p$ < .001) were associated with rehospitalization. Patients struggling to breath were 10.7 times more likely to be readmitted than those not struggling to breath, whereas patients who reported feeling well in general were less likely to be readmitted by a 55% margin. Those adherent with taking their medications had fewer breathing problems, reported feeling less tired and were less likely to be readmitted. Daily weight, blood pressure, and heart rate had no impact on readmission. Fig 2 presents the corresponding Kaplan-Meier readmission curve.

**Table 2. Univariate analysis using Cox model to identify baseline risk factors.**

| Parameter | Parameter Estimate | Standard Error | Hazard Ratio | 95% Hazard Ratio Confidence Limits | | Pr > ChiSq |
|---|---|---|---|---|---|---|
| Gender | 0.05409 | 0.63284 | 1.056 | 0.305 | 3.649 | 0.9319 |
| Ethnicity | 0.06146 | 0.63273 | 1.063 | 0.308 | 3.675 | 0.9226 |
| Marital Status | -0.57849 | 0.69059 | 0.561 | 0.145 | 2.171 | 0.4022 |
| History of HF | -0.84016 | 1.06354 | 0.432 | 0.054 | 3.471 | 0.4296 |
| Hypertension | 0.71114 | 0.79110 | 2.036 | 0.432 | 9.599 | 0.3687 |
| Diabetes Mellitus | 1.11870 | 0.69053 | 3.061 | 0.791 | 11.848 | 0.1052 |
| Coronary Heart Disease | 0.56184 | 0.67118 | 1.754 | 0.471 | 6.536 | 0.4025 |
| Valvular Heart Disease | -0.29567 | 0.73068 | 0.744 | 0.178 | 3.116 | 0.6857 |
| Atrial Fibrillation | 0.18693 | 0.67148 | 1.206 | 0.323 | 4.495 | 0.7807 |
| Chronic Kidney Disease | 0.32540 | 0.64613 | 1.385 | 0.390 | 4.913 | 0.6145 |
| Pulmonary Disease | 0.82703 | 0.73080 | 2.287 | 0.546 | 9.577 | 0.2578 |
| HF Etiology-Ischemic | 0.82002 | 0.80357 | 2.271 | 0.470 | 10.968 | 0.3075 |
| HF Etiology-Hypertension | -15.07341 | 3079 | 0.000 | 0.000 | . | 0.9961 |
| HF Etiology-Arrhythmias | 0.24630 | 1.06988 | 1.279 | 0.157 | 10.415 | 0.8179 |
| HF Etiology-Valvular | -14.06566 | 2091 | 0.000 | 0.000 | . | 0.9946 |
| Tobacco | 1.23039 | 0.69527 | 3.423 | 0.876 | 13.371 | 0.0768 |
| Alcohol | -1.63648 | 1.05611 | 0.195 | 0.025 | 1.543 | 0.1213 |
| ACEI or ARB | 0.56302 | 0.70876 | 1.756 | 0.438 | 7.044 | 0.4270 |
| Digoxin | -0.03742 | 1.05462 | 0.963 | 0.122 | 7.611 | 0.9717 |
| Aldactone | -1.07487 | 0.79213 | 0.341 | 0.072 | 1.612 | 0.1748 |
| Statin | 2.28754 | 1.05553 | 9.851 | 1.245 | 77.971 | 0.0302* |
| ASA | 0.49645 | 0.64648 | 1.643 | 0.463 | 5.833 | 0.4425 |
| Ambulation | 0.09525 | 1.05474 | 1.100 | 0.139 | 8.693 | 0.9280 |
| Bathing | 0.04916 | 1.05477 | 1.050 | 0.133 | 8.302 | 0.9628 |
| Toileting | 0.68369 | 1.05611 | 1.981 | 0.250 | 15.699 | 0.5174 |
| ADLs | 0.68369 | 1.05611 | 1.981 | 0.250 | 15.699 | 0.5174 |
| Use of Assist Devices | 0.09512 | 0.79086 | 1.100 | 0.233 | 5.182 | 0.9043 |
| Age | 0.00927 | 0.03177 | 1.009 | 0.948 | 1.074 | 0.7704 |
| BNP | -0.0000330 | 0.0000523 | 1.000 | 1.000 | 1.000 | 0.5282 |
| Sodium | -0.06773 | 0.05264 | 0.935 | 0.843 | 1.036 | 0.1982 |
| BUN | 0.01824 | 0.01490 | 1.018 | 0.989 | 1.049 | 0.2209 |
| Creatinine | 0.54019 | 0.44130 | 1.716 | 0.723 | 4.076 | 0.2209 |
| Hgb | -0.25817 | 0.19147 | 0.772 | 0.531 | 1.124 | 0.1776 |
| LVEF | -0.01417 | 0.02513 | 0.986 | 0.938 | 1.036 | 0.5757 |
| HR | 0.00200 | 0.01953 | 1.002 | 0.964 | 1.041 | 0.9182 |
| SBP | 0.01425 | 0.01490 | 1.014 | 0.985 | 1.044 | 0.3389 |
| DBP | -0.04260 | 0.04467 | 0.958 | 0.878 | 1.046 | 0.3403 |
| Admission Weight | 0.01331 | 0.01709 | 1.013 | 0.980 | 1.048 | 0.4360 |
| Discharge Weight | 0.01532 | 0.01639 | 1.015 | 0.983 | 1.049 | 0.3501 |
| BMI | 0.04875 | 0.05761 | 1.050 | 0.938 | 1.175 | 0.3974 |
| Intake and Output | 0.0002426 | 0.0003930 | 1.000 | 0.999 | 1.001 | 0.5370 |
| Lower Extremity Edema | -0.07570 | 0.64575 | 0.927 | 0.261 | 3.287 | 0.9067 |
| Dyspnea | 0.17600 | 0.20689 | 1.192 | 0.795 | 1.789 | 0.3949 |

**Note**: for categorical variable. the reference group for ethnicity is non-white, the refence group for marital status is married. For all other categorical variables, reference group is not shown in the table because the estimate is zero.

*p<0.05

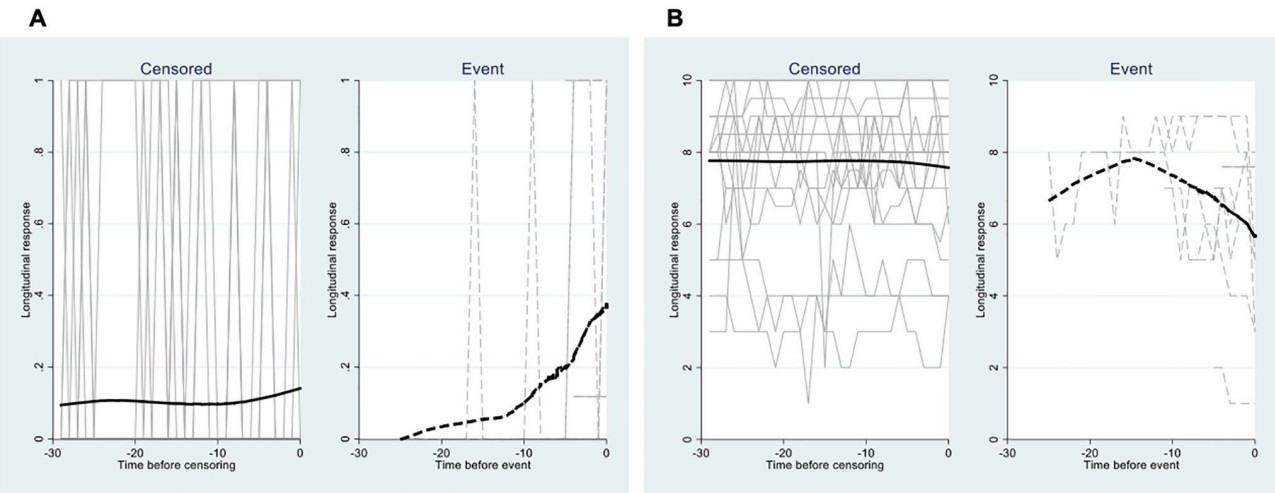

**Fig 1. Longitudinal trajectory plots for Q1 (A: Struggling to breath) and Q13 (B: How do you feel in general?).** Time scale adjusted by subtracting patient's time to readmission/censoring (0 indicates point of readmission or censoring). Q1 shows an exponential increase in *struggling to breath* as time to readmission draws closer. Q13 shows a linear decrease in *how patients feel* about 2 weeks before time to readmission. There was virtually no change for patients not readmitted (censored) for either question.

## Discussion

This study evaluated patient responses to daily symptomatic, physiologic, and adherence characteristics and the relationship of these experiences to 30-day HF readmissions. The study findings highlight the importance of patients' perceived symptom burden as patients who reported dyspnea and feeling unwell were more likely to be readmitted to the hospital within 30 days compared to patients without these symptoms. Previous studies have shown an association between readmissions in patients with HF and demographic, physiological, laboratory, etiology, functional class, comorbid conditions, clinical, and functional data [7–16, 19–22]. These relationships were not replicated in this study. This may be due to the small sample size in this study or specific characteristics in this sample population who were primarily obese African American women with non-ischemic cardiomyopathies resulting in HFrEF. This inconsistent finding may also represent incomplete symptom evaluations in prior studies.

Despite efforts to care for patients with HF out of hospital, both 30-day and 90-day readmission rates increased between 2010 and 2017 to 19.9% and 34.6% respectively, resulting in high healthcare utilization and poor patient quality of life [42]. Importantly, the overall 30-day readmission rate in this study was higher than previously reported at 32% despite more than 90% of participants being able to perform ADLs at hospital discharge, indicating baseline functional ability. In addition, most readmissions occurred during the initial two-week period after discharge.

### Symptomatic experience

There remains a fundamental gap in understanding how to engage patients with HF in symptom recognition and response. Our findings demonstrate that self-reported dyspnea, characterized as struggling to breath, was more associated with the likelihood of hospital readmission than the demographic, physiological, and laboratory parameters measured. Dyspnea, a highly subjective finding, recognized as a hallmark symptom of HF, may be difficult to quantify. Attempts have been made to quantify symptom burden in HF, although such efforts run the

**Table 3. Joint longitudinal and readmission models for patient-reported questions.**

| Question | Hazard Ratio | 95% CI | p value |
|---|---|---|---|
| 1. Struggling to breath? | 10.65 | 1.69–67.10 | 0.012* |
| 2. Difficulty concentrating? | 1.52 | 0.23–10.08 | 0.662 |
| 3. Weight | 0.99 | 0.97–1.01 | 0.513 |
| 4a. Systolic BP | 0.99 | 0.96–1.02 | 0.512 |
| 4b. Diastolic BP | 1.00 | 0.95–1.06 | 0.864 |
| 5. Heart rate | 1.02 | 0.95–1.10 | 0.545 |
| 6. Trouble breathing lying down? | 1.55 | 0.36–6.72 | 0.560 |
| 7. Difficulty breathing when walking? | 1.51 | 0.38–6.03 | 0.561 |
| 8. Swelling in feet? | 1.31 | 0.34–5.05 | 0.698 |
| 9. Take medications today? | 0.58 | 0.11–2.95 | 0.509 |
| 10. How is your breathing? | 0.77 | 0.51–1.16 | 0.205 |
| 11. How tired do you feel? | 0.82 | 0.58–1.16 | 0.259 |
| 12. How do you feel in general? | 0.45 | 0.27–0.70 | < 0.001* |

Hazard ratios adjusted for age and sex.

CI: confidence interval

BP: blood pressure

*p<0.05

risk of diminishing the value of patient report [43]. Despite potential reporting bias, patient reports of dyspnea were previously found to be more effective than pulmonary function testing in predicting chronic obstructive pulmonary disease (COPD) mortality [44]. Of equal import, self-reported general well-being (e.g. Likert scale "how do you feel in general?") also significantly (inversely) correlated with the likelihood of readmission. The single-item concept of general well-being–which may encompass physical, emotional, and/or spiritual aspects of the illness experience–while in itself not intended to replace more comprehensive survey inventories, may be a particularly helpful surrogate for symptom burden in advanced illness [45]. Congruent with this, patients include emotional reactions as a component of HF self-care [46].

The findings of this study conclude that the symptom cluster of struggling to breath and generally feeling unwell coincided with help-seeking behavior. These findings are consistent with the SMT premise that symptom perception is a requisite to self-care and persistent symptoms that are perceived as distressing result in help-seeking responses and ultimately outcomes [27–29]. Patients develop strategies for HF self-care which are formed from adaptations and integration from their previous cumulative experiences [46]. Patients with HF that demonstrate more effective self-care practices have better quality of life and lower mortality and readmission rates than those with lower levels of self-care [47, 48].

Well established guidelines for post-hospital care transitions include a post-discharge follow-up phone call within 3 days of discharge and an outpatient follow up visit within 7 days of discharge [49–51]. Although these visits are often conducted, the focus includes objective indicators of clinical stability rather than intentional inquiry into the patient's symptomatic experience to elicit important indicators that portend future readmissions. It is noteworthy that the onset of patient perceived dyspnea and feeling unwell occurred approximately two weeks prior to readmission when the symptoms began to manifest and increase in frequency and intensity. This study finding is congruent with the timing of thoracic impedance changes associated with hospitalization [52].

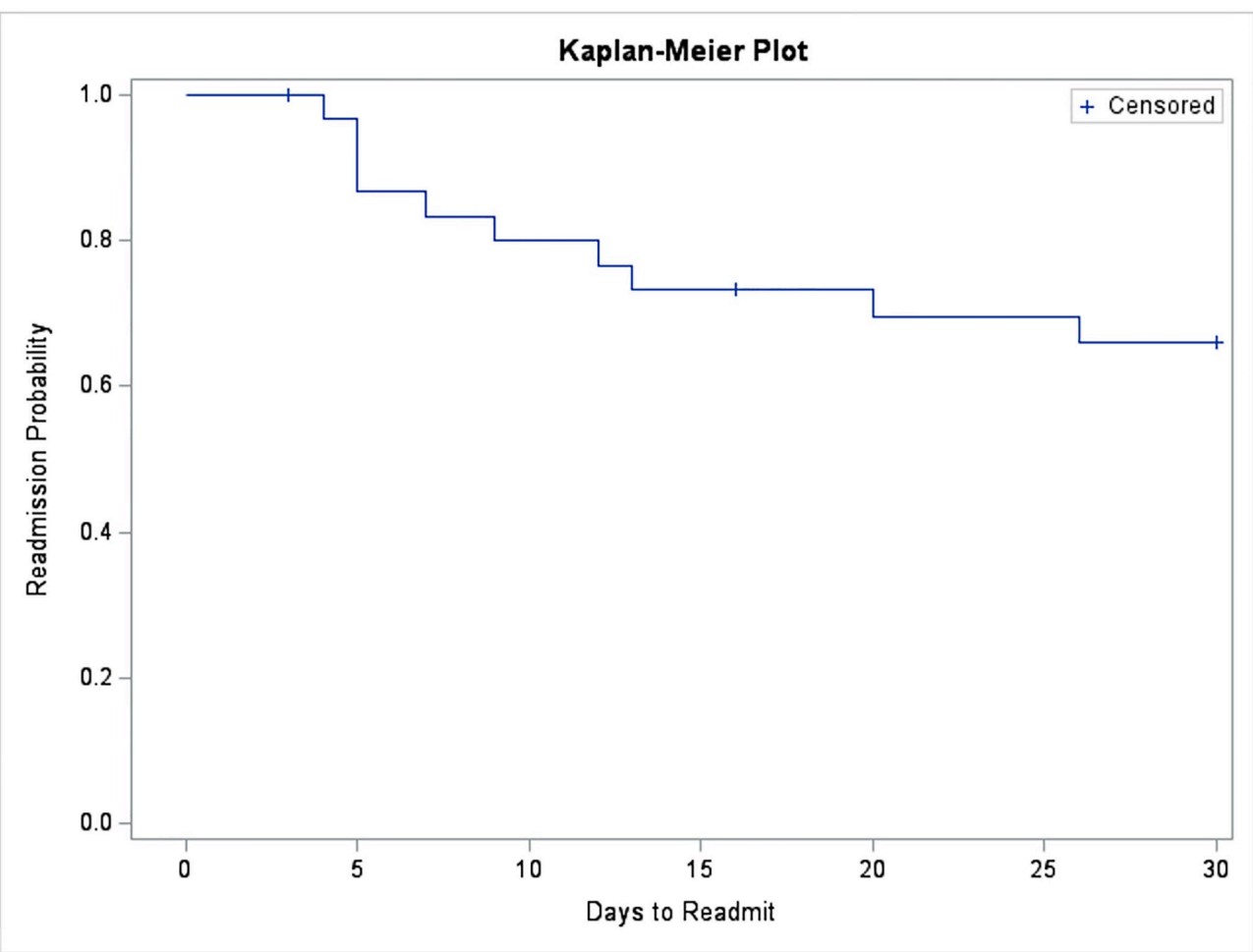

**Fig 2. Kaplan Meier plot of readmission probability.**

Studies suggest that patient-cardiologist concordance in symptom reporting correlates with symptom prevalence–less commonly reported symptoms, such as dyspnea at rest or dyspnea when lying flat, are underreported by cardiologists, perhaps in turn delaying necessary treatment modification [53]. It is important for clinicians and patients to attune to and appreciate the role of nuanced symptom assessment and reporting in the illness experience, emphasizing even subtle changes as potential hallmarks of disease instability. Our study underlines how patient perception of struggling to breath and generally feeling unwell may herald consequential changes in weight and physiologic alternations associated with unplanned health care encounters. Emphasis on symptom assessment and reporting may lead to earlier health seeking behaviors that may confer benefits of interventions and better outcomes, as symptoms remain the main driver of health-related outcomes for patients with HF, including hospitalizations.

## Daily weight

The role of patient education regarding daily weight monitoring and reporting is primarily based on expert consensus. Despite the lack of supportive evidence or association with

readmissions, daily weights remain a cornerstone of HF patient self-monitoring. The AHA Self-Check Plan for HF Management recommends that patients notify a healthcare professional if there is an increase of $\geq$ 2–3 pounds in 24 hours or $\geq$ 5 pounds in one week [33]. The most recent update of HF management guidelines continues to recommend patient monitoring of weight fluctuations as a Class I recommendation, and the 2021 ACC/AHA Key Data Elements for HF advise documentation of patient confirmation of daily self-monitoring of weight [38, 51]. Conflicting data on the effect of weight fluctuation in predicting HF rehospitalization come largely from older studies. A more recent post hoc analysis of the ASCEND-HF (Acute Study of Clinical Effectiveness of Neseritide and Decompensated Heart Failure) trial found that patients who experienced weight gain following hospital discharge had higher readmission rates and reduced survival [54]. By contrast, our study did not demonstrate an association between daily weights, weight trends, or weight changes on 30-day rehospitalizations.

## Diversity

Despite similar, poor outcomes among genders, women experience HF differently than men including older age of onset, increased prevalence of HFpEF, and a greater symptom burden in HFrEF [55]. However, women have been underrepresented in HF clinical studies. In this study, females comprised 54% of the participants. Socioeconomic risk factors associated with HF readmissions include non-white race, income, and payment source [6]. Diverse populations have been challenging to represent in HF studies; however, this study included a majority of African American participants at 63%. Although mean income was not a component of demographic data collected in this study, the study institution serves a disadvantaged population. An incidental finding was that the majority of those enrolled required the provision of a weight scale and blood pressure cuff which were necessary for study participation and were provided as needed. This finding highlights additional potential challenges for HF patients who may not have the resources necessary to effectively self-manage their condition.

## Statin

The only baseline characteristic appearing to have an impact on hospital readmission was the presence of a statin medication at discharge. The paradoxical role of statins in HF management has been documented. Statins are known to reduce the risk of HF in those with ischemic heart disease, however, low total serum cholesterol has been associated with poor prognosis in patients with HF [56–61]. The effects of statin therapy in patients with existing HF on HF hospital admissions is also unclear due to conflicting trial results and trial limitations [59–64]. There are many proposed mechanisms for the potential negative effects of statin therapy on HF readmissions including inhibition of CoQ10 synthesis [65] and prosarcopenic effects [66]. Studies are ongoing.

## Clinical practice recommendations

Based on the findings from this study, clinical practice recommendations include asking patients about their general well-being, and symptomatic experiences. Providers and patients may benefit from appreciating the nuances of early symptomatic changes as predictors of decompensation, rather than delaying until the development of acute symptomatic manifestations. As more HF management is taking place in the community and home settings, it is crucial to include self-monitoring as a part of integrative HF care and management. In addition to the customary logging of weight, pulse, and blood pressure changes, daily self-management should focus on daily monitoring of the symptomatic experience of patients with HF, and promoting medication adherence.

Drawing from these study findings, we confirmed that those patients who reported adherence with medications had fewer breathing problems, reported feeling less tired and were less likely to be readmitted.

Patients who are hospitalized for HF have experienced significant symptomatic decline and an acute decompensation of this chronic disease. Patients with HF often benefit from palliative care for the management of the highly prevalent symptomatic experience, a hallmark of this disease, in addition to contemporary guideline-based care with HF disease modifying therapies [67, 68]. Our study underscores the complexity of care that HF patients face daily including the management of intricate medication regimens, challenging comorbidities such as diabetes mellitus and chronic kidney disease, and burdensome manifestations such as dyspnea and fluid retention.

Moreover, patients' reported symptoms should be monitored and promptly evaluated by interdisciplinary team members to prevent exacerbations that could necessitate hospital readmission. Vigilant interdisciplinary management of complex comorbid conditions is also necessary to ensure optimal post-discharge outcomes. Future HF studies investigating self-reported symptom burden and disease progression and/or risk of unplanned health care encounters should include measures specific to dyspnea and general well-being. Integrating such assessments into routine post-hospitalization care reflects an important reality that patients have their own unique illness experiences, regardless of what objective measurements may indicate.

Findings from this study contribute to future research by identifying key predictors of 30-day HF post-discharge outcomes via self-reported patient symptoms. The findings of this study advance our understanding of the effects of patient self-monitoring strategies and patient-reported symptoms after HF hospitalization by clarifying the strength of these indicators in predicting 30-day patient outcomes and readmissions following hospital discharge. Additionally, this study contributed to modeling the significant predictors of clinical stability in the home environment following discharge for acute decompensated HF and provided important, valid and timely information about the evaluation of HF patients and risk of adverse post-hospitalization outcomes.

## Limitations

While the inclusion and exclusion criteria were followed, confounding factors such as inpatient care practices, medication dosing, outpatient management, and individual patient differences may have affected the outcomes in this complex patient population. Moreover, the small sample size, observational design, and mostly African American participants limits the statistical approaches and the generalizability of findings to other HF populations. In addition, there was a high participant attrition rate, and missing data that required statistical adjustments.

It is also important to note that 37% of study participants had pulmonary disease. Reported shortness of breath in these patients could be, at least partially, attributable to this co-occurring disease process. Thus, it is difficult to determine if hospital readmissions occurred solely due to HF exacerbation. Moreover, additional factors known to impact HF patient outcomes including cognitive impairment, depression, anxiety and health literacy, were not considered in this study.

## Future research

Future studies are required to understand the biophysical and psychological factors involved in the complex concept of perceived shortness of breath and feeling unwell to equip patients to become experts in symptom identification to promote early intervention and improve outcomes. Conducting multicenter studies and expanding the sample size will allow evaluation of

the types and intensities of HF symptoms that portend poor outcomes, and be more generalizable across HF populations. We may require a reframing of our definition of significant symptoms for provider identification and patient teaching. Furthermore, providing effective patient education, and developing individualized self-monitoring plans may help prevent patients from unnecessary future hospital readmissions. Evaluating self-care outcomes are needed to facilitate patient actions based on responses to symptoms once they are recognized, and to meaningfully empower patients with symptom response behaviors.

Additionally, investigations of mechanisms, causes, and possible therapies unique to women with HF could be clinically valuable. Large randomized controlled trials are needed to evaluate the effects of stain therapy on those with chronic HF in terms of mortality and morbidity including hospitalization rates. The clinical implications of clarifying these associations are immense.

## Conclusion

Patient-perceived symptoms, particularly dyspnea, characterized as struggling to breath, and feeling unwell were more important factors than the demographic, physiological, or laboratory parameters in predicting 30-day HF re-hospitalizations after discharge for the period of 30 days.

## Supporting information

**S1 Data. Minimal data set.**
(XLSX)

## Author Contributions

**Conceptualization:** Kelley M. Anderson.

**Formal analysis:** Paul Kolm, Haijun Wang.

**Funding acquisition:** Kelley M. Anderson.

**Investigation:** Kelley M. Anderson.

**Methodology:** Kelley M. Anderson.

**Supervision:** Kelley M. Anderson.

**Visualization:** Paul Kolm, Haijun Wang.

**Writing – original draft:** Kelley M. Anderson, Dottie Murphy, Vera Barton-Maxwel.

**Writing – review & editing:** Kelley M. Anderson, Dottie Murphy, Hunter Groninger, Vera Barton-Maxwel.

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
