## [Decision Letter · Decision Letter 0]

17 Mar 2022

PONE-D-22-05107Perceived Symptoms as the Primary Indicators for 30-Day Heart Failure ReadmissionPLOS ONE

Dear Dr. Anderson,

Thank you for submitting your manuscript to PLOS ONE. After careful consideration, we feel that it has merit but does not fully meet PLOS ONE’s publication criteria as it currently stands. Therefore, we invite you to submit a revised version of the manuscript that addresses the points raised during the review process.

We look forward to receiving your revised manuscript.

Kind regards,

Tariq Jamal Siddiqi

Academic Editor

PLOS ONE

Journal Requirements:

4. Please ensure that you include a title page within your main document. You should list all authors and all affiliations as per our author instructions and clearly indicate the corresponding author.

Reviewers' comments:

Reviewer's Responses to Questions

**Comments to the Author**

1. Is the manuscript technically sound, and do the data support the conclusions?

Reviewer #1: Partly

2. Has the statistical analysis been performed appropriately and rigorously? 

Reviewer #1: Yes

3. Have the authors made all data underlying the findings in their manuscript fully available?

Reviewer #1: Yes

4. Is the manuscript presented in an intelligible fashion and written in standard English?

Reviewer #1: Yes

5. Review Comments to the Author

Reviewer #1: Anderson et al. conducted a study on “Perceived symptoms as the primary indicators for 30-Day heart failure readmission”, in which they explored the association between clinical indications, physiological measures and self-reported symptoms post discharge, and rehospitalization in heart failure patients. They found a significant association of rehospitalization in patients reporting trouble breathing and feeling unwell. Statin use was also associated with rehospitalization. No other clinically indications, physiological measures or self-reported symptoms were associated.

In my opinion, this study may be improved by incorporating the following edits:

1. The gap in the existing literature, that this study seeks to fill, has not been appropriately highlighted. This should be done in the introduction section, to justify the need to conduct this study. Whilst sufficient background on the topic has been provided, the authors should clearly state whether studies similar to this one, evaluating self-reported symptoms, have been published, or if this is a fundamentally novel idea or implementation.

2. Clearly highlighting statistically significant outcomes in the tables, perhaps with an asterisk, would significantly improve readability and comprehensiveness of the data.

3. A clarifying statement about the spread of patients between HFrEF and HFpEF in this study would be appreciated. A stratification and comparison of data on these grounds or compelling reason against such an analysis could also be considered.

4. In lines 191-194, please remove the statements prefaced with the phrase “although not statistically significant”. Statistical non-significance fundamentally indicates the absence of compelling data to this end. Claiming a “higher hazard of readmission”, despite statistical non-significance, is erroneous.

5. In in lines 214-219, the authors highlight relationships which have been associated with rehospitalization in previous studies but were not demonstrably prominent in these results. Please consider adding a statement also suggesting that these findings may be a results of the small sample size of this study and that further research and data would be needed to this end to provide a definitive conclusion.

6. In lines 252-253 the authors state that “this study affirmed that those patients who reported adherence with medications had fewer breathing problems, reported feeling less tired and were less likely to be readmitted.” This finding, as presented in Table 3 and Q9, presented with statistical non-significance. As mentioned above, providing statements about increased or decreased likelihood despite statistical non-significance is erroneous. Please remove this statement.

6. PLOS authors have the option to publish the peer review history of their article (what does this mean?). If published, this will include your full peer review and any attached files.

Reviewer #1: **Yes: **Muhammad Talha Maniya

---

## [Author Response · Author response to Decision Letter 0]

6 Apr 2022

Dear Editor and Reviewers, 

Thank you kindly for your thorough review of the manuscript and for your comments and suggestions. We have revised the manuscript and believe it is strengthened after the opportunity to address your recommendations. We hope that our revisions are clear and acceptable to you. Thank you again.

From Editor Response to Editor

Thank you, we have reviewed these resources and updated accordingly. We have also updated the reference numbers in the text to the format of brackets [1].

A title page was added at the beginning of the main document, the title page lists all authors and affiliations.

The minimal data set was obtained by the study statistician and will be uploaded as a Supporting Information file. The file is fully anonymized.

4. Please ensure that you include a title page within your main document. You should list all authors and all affiliations as per our author instructions and clearly indicate the corresponding author. 

A title page was added at the beginning of the main document, the title page lists all authors and affiliations, and clearly indicates the corresponding author.

From Reviewer #1 

1. The gap in the existing literature, that this study seeks to fill, has not been appropriately highlighted. This should be done in the introduction section, to justify the need to conduct this study. Whilst sufficient background on the topic has been provided, the authors should clearly state whether studies similar to this one, evaluating self-reported symptoms, have been published, or if this is a fundamentally novel idea or implementation. 

Yes, thank you for pointing this out. We have revised the last two sentences of the introduction 

From: “Therefore, further research is critical to evaluate the post-hospital experience of patients with HF and the symptomatic characteristics related to rehospitalizations. This study aimed to evaluate patient responses to daily symptomatic, physiologic, and medication adherence characteristics and the relationship of these patient-centric experiences to 30-day HF post-hospitalization readmissions.”

To: “However, studies evaluating the subjective post-hospital experience of patient-reported symptoms to predict HF readmissions are lacking. This study addresses this gap by investigating daily symptomatic, physiologic, and medication adherence characteristics and the relationship of these patient-centric experiences to expand our understanding of factors contributing to 30-day HF post-hospitalization readmissions.”

2. Clearly highlighting statistically significant outcomes in the tables, perhaps with an asterisk, would significantly improve readability and comprehensiveness of the data. 

Statistically significant outcomes were highlighted on the tables with an asterisk.

3. A clarifying statement about the spread of patients between HFrEF and HFpEF in this study would be appreciated. A stratification and comparison of data on these grounds or compelling reason against such an analysis could also be considered. 

This is an important question, and based on our anecdotal experiences caring for patients we would agree that there are likely differences in the symptomatic experiences between HFrEF and HFpEF populations, for example more fatigue in HFpEF. 

From the statistician: LVEF was included in the analyses as a continuous variable and was not statistically significantly related to any of the outcomes. Dichotomizing as HFrEF and HFpEF would not result in anything different. For example, comparing the two groups with respect to readmission within 30 days was not significant, p = 0.507.

This study also was not powered for specific sub-group analysis and there were uneven numbers in the HFrEF and HFpEF groups. We did have a robust discussion about this point and do believe this is an important consideration for future research with an intentional study design to answer the question. 

4. In lines 191-194, please remove the statements prefaced with the phrase “although not statistically significant”. Statistical non-significance fundamentally indicates the absence of compelling data to this end. Claiming a “higher hazard of readmission”, despite statistical non-significance, is erroneous. 

Understood, this sentence has been deleted.

5. In lines 214-219, the authors highlight relationships which have been associated with rehospitalization in previous studies but were not demonstrably prominent in these results. Please consider adding a statement also suggesting that these findings may be a results of the small sample size of this study and that further research and data would be needed to this end to provide a definitive conclusion.

This was revised,

From: “This may represent incomplete symptom evaluations in the prior studies, or some specific characteristics in this sample population who were primarily obese African American women with non-ischemic cardiomyopathies resulting in HFrEF.”

To: “This may be due to the small sample size in this study or specific characteristics in this sample population who were primarily obese African American women with non-ischemic cardiomyopathies resulting in HFrEF. This inconsistent finding may also represent incomplete symptom evaluations in prior studies.”

We did include the small sample size in the limitations section and the section of future research.

6. In lines 252-253 the authors state that “this study affirmed that those patients who reported adherence with medications had fewer breathing problems, reported feeling less tired and were less likely to be readmitted.” This finding, as presented in Table 3 and Q9, presented with statistical non-significance. As mentioned above, providing statements about increased or decreased likelihood despite statistical non-significance is erroneous. Please remove this statement. 

This statement was removed.

---

## [Editor Report · Decision Letter 1]

18 Apr 2022

Perceived Symptoms as the Primary Indicators for 30-Day Heart Failure Readmission

PONE-D-22-05107R1

Dear Dr. Anderson,

We’re pleased to inform you that your manuscript has been judged scientifically suitable for publication and will be formally accepted for publication once it meets all outstanding technical requirements.

Kind regards,

Tariq Jamal Siddiqi

Academic Editor

PLOS ONE
---

## [Editor Report · Acceptance letter]

20 Apr 2022

PONE-D-22-05107R1 

Perceived Symptoms as the Primary Indicators for 30-Day Heart Failure Readmission 

Dear Dr. Anderson:

I'm pleased to inform you that your manuscript has been deemed suitable for publication in PLOS ONE. Congratulations! Your manuscript is now with our production department. 

Kind regards, 

on behalf of

Dr. Tariq Jamal Siddiqi 

Academic Editor

PLOS ONE